

# Contents and yields of copper, iron, manganese and zinc would be affected by lucerne age and cut

Zhennan Wang[1], Yizhao Shen[2], Chongliang Bi[1], Mirielle Pauline[3], Qingping Zhang[1], Shenjin Lv[1], Huimin Yang[4] and Yan Yang[5]

[1] College of Agriculture and Forestry Sciences, Linyi University, Linyi, China
[2] College of Animal Science and Technology, Hebei Agricultural University, Baoding, China
[3] Department of Pediatrics, University of Alberta, Edmonton, Canada
[4] State Key Laboratory of Grassland Agro-ecosystems, College of Pastoral Agriculture Science and Technology, Lanzhou University, Lanzhou, China
[5] Linyi Academy of Agricultural Science, Linyi, China

Corresponding authors
Zhennan Wang,
wangzhennan@lyu.edu.cn
Huimin Yang, huimyang@lzu.edu.cn

## ABSTRACT

**Background:** Lucerne is a perennial legume forage, which can produce multiple cuts in 1 year. Microelements play fundamental roles in the function, maintenance and adaptation to the environment for lucerne growth. However, the role of the accumulation of copper (Cu), iron (Fe), manganese (Mn) and Zinc (Zn), which vary with lucerne ages or cuts, has not been previously determined. Therefore, a hypothesis on the Cu, Fe, Mn and Zn in lucerne varying with age and cut was tested.
**Methods:** A total of 11, 8, 5, 4 and 1 year old lucerne (*Medicago sativa* Longdong) were selected as the material (until 2012 year), and samples were taken as three cuts at the cutting periods (early flowering stage) in 2012. Then, the contents and yields of Cu, Fe, Mn and Zn in lucerne were measured and calculated.
**Results:** The highest contents of Cu, Fe, Mn and Zn in lucerne were found in the 1 year old among the five ages, at the 3rd cut compared to the other two cuts, and in the leaf among the three organs. The highest yields of Cu, Fe, Mn and Zn were found in the older ages (11 and 8 years old), at the 3rd cut, and in the root among the three organs. The most positive correlations were found between contents, yields and biomass.
**Conclusions:** The hypothesis was supported by the results. And the contents and yields of lucerne Cu, Fe, Mn and Zn were affected by the age, cut and organ. Furthermore, the yields of lucerne Cu, Fe, Mn and Zn were determined by their contents and lucerne biomass.

## INTRODUCTION

Nutrient elements in plants play a basic role during their growth and adaptation to the environment (*Wright & Westoby, 2003*; *Imaran & Gurmani, 2011*; *Kováčik & Škarpa, 2019*). Generally, carbon content increases (*Marković et al., 2009*), and mineral nutrient (macroelements & microelements) contents are gradually diluted as the plant grows

(*Tyrolová & Výborná, 2008*; *Marković et al., 2009*; *Yang & Luo, 2011*). But their accumulations would increase as the plant biomass increased (*Hooker & Compton, 2003*). These changes were generally observed in some annual plants or in some perennial plants within one cut or a growing season. However, the element contents change more complexly due to a longer lifetime and frequent use annually for the perennial plant (*Wang et al., 2015*; *Akburak, 2020*). Thus, exploring how element contents change with plant age or growing season are important to better understand the adaptation of a perennial plants to the environment.

Similar to macroelements, microelements are also needed for plant growth (*Marković et al., 2009*; *Lukin et al., 2018*). They are the components and prosthetic groups of many enzymes, and would lead to a high plant yield and high chemical constituents (such as carbohydrates, protein, oil) (*EISayed et al., 2018*; *Afsahi et al., 2020*; *Ierna et al., 2020*). Adversely, when plants lack a microelement, young leaves and stems will have limited growth with yellowing leaves, the plant withering, the leaves falling off, and the leaves having mottled spots (*Imaran & Gurmani, 2011*; *Lu et al., 2020*). Furthermore, the nutrient-deficient plant as forage are fed to livestock which could experience a range of dietary mineral imbalances unless supplementary mineral sources are provided (*Raeside, Nie & Behrendt, 2012*; *Kletikova et al., 2020*). Correctly aligned phase nutrition program minimizes metabolic problems of livestocks (*Skalicka et al., 2016*). Livestock also choose initiatively to eat the fodder with the highest concentrations of plant microelements (such as Cu and Zn) for their advantage (*Azuma, Tomioka & Takenaka, 2016*). Thus, neglecting the microelements in plants not only negatively limits plant growth, but also negatively affects the health of livestock.

Lucerne is a kind of perennial legume forage with an average lifespan that exceeds 20 years (*Bennett, 2012*), has been cultivated almost 32 million hectare in the world and 1,300 thousands hectare in China, and it is continually cultivating more in recent years. It was domesticated as early as 7000 BCE, and it originated from Central Aisa (*Torabi et al., 2011*). In China, lucerne has more than 2,000 years of planting history. It is used as widespread forage with a high nutritional (include protein and amino acid et al.) (*Caunii et al., 2012*), economic and environmental protection values (*Li & Huang, 2008*). But the forage yield of lucerne decreases with increasing age, and *Li & Huang (2008)* suggested that lucerne should be utilized at more than 8 years old which was estimated by the comprehensive results of forage yield and soil water. *Wang et al. (2015)* has also suggested applying optimal N fertilization to the stands of 8 years or older lucerne for the persistence of lucerne production which was estimated by the macroelements (i.e., N and P). The age-effect of microelements is often neglected.

In this study, we tested four microelements, i.e., copper (Cu), iron (Fe), manganese (Mn) and zinc (Zn). The hypothesis was that Cu, Fe, Mn and Zn could change with stand age and cut of lucerne. The specific objectives were to find out: (1) how the contents of Cu, Zn, Mn and Zn differ among stand ages and cuts of lucerne; and (2) what are the relationships of the contents and yields of the four microelements in lucerne.

## MATERIALS AND METHODS

### Study site and climate data

The lucerne fields were located at the Qingyang station on the Loess Plateau, China. It is 107°51′E and 35°39′N. The elevation is 1,298 m. Mean annual precipitation (MAP) is between 480~660 mm, and 70% of MAP majorly falls from July to September. Mean annual temperature (MAT) is 8~10 °C, and the highest temperature is 39.6 °C, the lowest temperature is −22.4 °C. This area belongs to the rainfed area, and the climate has typical continental characteristics. The majority of the area's soil is Heilu soil (*Wang et al., 2015*).

### Material

The 11 year old (cultivated in 2002), 8 year old (cultivated in 2005), 5 year old (cultivated in 2008), 4 year old (cultivated in 2009) and 1 year old (cultivated in 2012) samples of lucerne (*Medicago sativa* Longdong) were selected as the material (until 2012 year). Samples were taken as three cuts (label as 1st, 2nd and 3rd cuts) at the cutting periods (early flowering stage) in 2012, the dates of sampling were June 6, August 12 and October 24. Then the samples were separated into leaf, stem and root. And the roots were only collected in the soil layer of 0~30 cm. Next the samples were oven-dried at 105 °C for 10 min and then at 80 °C for at least 48 h to a constant mass. At this time, the weight of all samples was recorded as biomass. Lastly the samples were smashed into uniform fine powder by a plant-sample mill and sieve mill with a 1 mm mesh for further measurements. All samples were repeated three times.

### Chemical analysis

The samples, which were sieve milled with a 1 mm mesh, were weighed to 0.2 g, and added into a digestion tube with 10 mL 1.4 g/cm$^3$ concentrated nitric acid (HNO$_3$). Then, the digestion tube was put into a microwave digestion instrument (MARS 240/50, USA), and digested for 1 h. The digestive liquid was put with deionized water into the beaker which was on a 220 °C hot plate, the volume was about 70 ml in the beaker. Once the liquid was less than 50 ml, the beaker was taken off the hot plate and cooled down naturally. The sample solution was transferred into a 100 ml volumetric flask, and filled to the scale line of 100 ml. Then, all samples were analyzed for Cu, Fe, Mn and Zn by an atomic absorption spectrometer (iCE 3500; Thermo fisher, Waltham, MA, USA) with an air-acetylene flame (FA-AAS) at a temperature of 2,300 °C (*Khan et al., 2011*; *Żurawik, Jadczak & Żurawik, 2013*; *Azuma, Tomioka & Takenaka, 2016*; *Li et al., 2020*). The yields of microelements were calculated as multiplying lucerne biomass by then contents of microelements.

### Data analysis

Multivariate analysis of variance was performed to assess the effects of cut, age, organ and their interactions on the contents of Cu, Fe, Mn and Zn in lucerne. Two-way analysis of variance was performed to assess the effects of cut, age and their interaction on the yields of Cu, Fe, Mn and Zn in lucerne. Differences among the five ages, three cuts or three

**Table 1 Effects (*F*-value) of cut, age, organ and their interactions on the contents (mg/kg) of Cu, Fe, Mn and Zn in lucerne.**

| Treatment | df | Cu | Fe | Mn | Zn |
|---|---|---|---|---|---|
| Cut | 2 | 41.793 ($P < 0.001$) | 29.133 ($P < 0.001$) | 52.622 ($P < 0.001$) | 45.489 ($P < 0.001$) |
| Age | 4 | 30.411 ($P < 0.001$) | 3.198 ($P = 0.05$) | 4.819 ($P = 0.002$) | 4.577 ($P = 0.002$) |
| Organ | 2 | 9.109 ($P < 0.001$) | 27.023 ($P < 0.001$) | 209.250 ($P < 0.001$) | 30.605 ($P < 0.001$) |
| Cut × Age | 7 | 88.180 ($P < 0.001$) | 3.278 ($P = 0.004$) | 2.921 ($P = 0.009$) | 6.715 ($P < 0.001$) |
| Cut × Organ | 4 | 9.132 ($P < 0.001$) | 3.805 ($P = 0.007$) | 8.724 ($P < 0.001$) | 0.807 ($P = 0.524$) |
| Age × Organ | 8 | 1.340 ($P = 0.235$) | 3.703 ($P = 0.001$) | 1.901 ($P = 0.070$) | 3.090 ($P = 0.004$) |
| Cut × Age × Organ | 14 | 2.492 ($P = 0.005$) | 2.660 ($P = 0.003$) | 2.042 ($P = 0.024$) | 1.985 ($P = 0.029$) |

Note:
*F*-values (*P*-values) were shown as the results. It was determined by multivariate analysis of variance. The significant difference was at $P < 0.05$.

**Table 2 Effects (*F*-value) of cut, age and their interaction on the yields (mg/m²) of Cu, Fe, Mn and Zn in lucerne.**

| Treatment | df | Biomass | Cu | Fe | Mn | Zn |
|---|---|---|---|---|---|---|
| Cut | 2 | 20.109 ($P < 0.001$) | 4.155 ($P = 0.026$) | 31.835 ($P < 0.001$) | 30.898 ($P < 0.001$) | 43.759 ($P < 0.001$) |
| Age | 4 | 12.946 ($P < 0.001$) | 2.912 ($P = 0.039$) | 4.923 ($P = 0.004$) | 7.029 ($P < 0.001$) | 10.101 ($P < 0.001$) |
| Cut × Age | 7 | 2.470 ($P = 0.042$) | 4.488 ($P = 0.002$) | 1.678 ($P = 0.155$) | 2.409 ($P = 0.046$) | 6.045 ($P < 0.001$) |

Note:
*F*-values (*P*-values) were shown as the results. It was determined by two-way analysis of variance. The significant difference was at $P < 0.05$.

organs were determined using one-way analysis of variance (ANOVA) and the least significant differences. The liner correlations of the lucerne contents and lucerne yields were analyzed with the model $y = ax + b$. Statistical analyses were performed using SPSS 17.0.

# RESULTS

## Effects of cut, age and organ on the contents and yields of Cu, Fe, Mn and Zn in lucerne

According to the multivariate analysis, the Cu, Fe, Mn and Zn contents of lucerne varied significantly as the effects of cut, age, organ and their interactions ($P < 0.05$, Table 1), except the Cu and Mn contents under the effect of age × organ, and the Zn content under the effect of cut × organ.

By using two-way analysis, there were significant differences on the Cu, Fe, Mn and Zn yields of lucerne at the effects of cut, age and their interaction ($P < 0.05$, Table 2), except the Fe yield under the effect of cut × age. The biomass of whole lucerne plants were also affected significantly by cut, age and their interaction ($P < 0.05$, Table 2).

## Cut-effect on the contents and yields of Cu, Fe, Mn and Zn in whole lucerne

The biomass of the whole plant increased with the increased cut. And the biomass of the 3rd cut was significantly higher than the biomass of the 1st and 2nd cuts ($P < 0.05$, Fig. 1A).

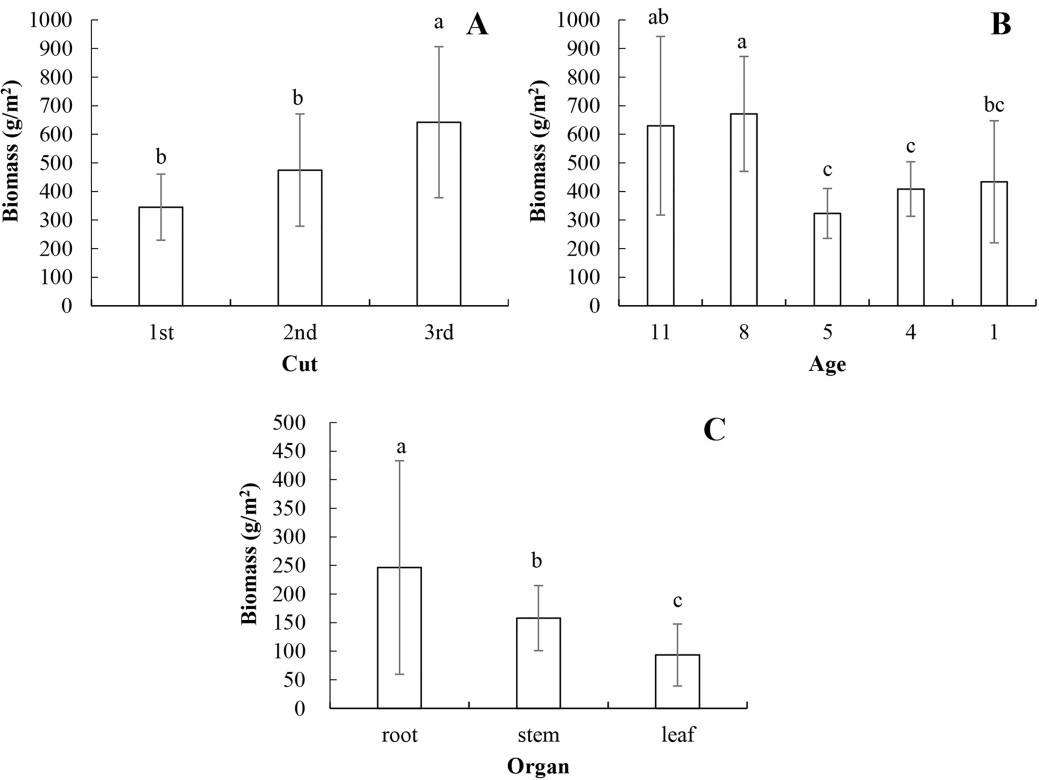

**Figure 1 The lucerne biomass on cut-effect (A), age-effect (B) and organ-effect (C).** The lucerne biomass on cut-effect and age-effect were analyzed with the whole plant. The error bars were the standard deviations. The significant difference was determined by one-way ANOVA, and it was represented as the different lowercase letters.

**Table 3 The contents and yields of Cu, Fe, Mn and Zn in the three cuts of whole lucerne.** The average ± standard deviation was shown as the results.

|  | Cut | Cu | Fe | Mn | Zn |
|---|---|---|---|---|---|
| Content (mg/kg) | 1st | 15.52 ± 4.30a | 600 ± 139b | 30.20 ± 5.22b | 32.19 ± 6.25a |
|  | 2nd | 13.08 ± 1.14a | 559 ± 184b | 33.60 ± 7.150b | 23.02 ± 3.82b |
|  | 3rd | 14.13 ± 6.07a | 1,026 ± 213a | 46.27 ± 5.73a | 37.50 ± 10.46a |
| Yield (mg/m$^2$) | 1st | 5.65 ± 2.94B | 211 ± 103B | 10.34 ± 3.98B | 11.62 ± 6.08B |
|  | 2nd | 6.18 ± 2.61AB | 271 ± 164B | 16.02 ± 8.55B | 10.71 ± 4.57B |
|  | 3rd | 8.04 ± 3.12A | 637 ± 260A | 29.29 ± 11.81A | 22.95 ± 8.26A |

**Note:**
The significant difference was determined by one-way ANOVA, it was represented as the different lowercase letters on the contents, and as the different capital letters on the yields (P<0.05).

There was no significant difference of Cu content of the whole plant on the cut-effect ($P > 0.05$, Table 3). The lucerne Fe and Mn contents of 3rd cut were significantly higher than the other cuts ($P < 0.05$). And lucerne Zn content of the 2nd cut were significantly lower than the other cuts. The yields of Cu, Fe, Mn and Zn increased significantly with the increased cuts ($P < 0.05$).

**Table 4 The contents and yields of Cu, Fe, Mn and Zn in the five ages of whole lucerne.** The average ± standard deviation was shown as the results.

|  | Age | Cu | Fe | Mn | Zn |
|---|---|---|---|---|---|
| Content (mg/kg) | 11 | 12.57 ± 3.48a | 748 ± 256a | 38.71 ± 8.22ab | 30.12 ± 4.23b |
|  | 8 | 12.66 ± 4.16a | 645 ± 185a | 32.11 ± 7.93b | 27.52 ± 8.89b |
|  | 5 | 14.56 ± 4.94a | 783 ± 426a | 37.73 ± 7.69ab | 29.08 ± 6.32b |
|  | 4 | 14.73 ± 3.82a | 681 ± 292a | 34.98 ± 11.02b | 30.28 ± 7.80b |
|  | 1 | 17.26 ± 4.99a | 879 ± 137a | 44.79 ± 8.45a | 40.18 ± 17.50a |
| Yield (mg/m$^2$) | 11 | 7.24 ± 3.10A | 532 ± 397A | 26.44 ± 16.66A | 19.26 ± 10.74A |
|  | 8 | 7.87 ± 1.52A | 448 ± 221A | 22.38 ± 10.93AB | 18.16 ± 7.02A |
|  | 5 | 4.94 ± 2.49A | 258 ± 176A | 12.25 ± 4.32B | 9.53 ± 3.76B |
|  | 4 | 5.86 ± 1.36A | 294 ± 188A | 14.90 ± 7.78AB | 12.36 ± 4.19AB |
|  | 1 | 8.01 ± 5.50A | 395 ± 253A | 20.02 ± 12.61AB | 18.45 ± 12.55A |

Note:
The significant difference was determined by one-way ANOVA, it was represented as the different lowercase letters on the contents, and as the different capital letters on the yields ($P < 0.05$).

**Table 5 The contents and yields of Cu, Fe, Mn and Zn in the three organs of lucerne.** The average ± standard deviation was shown as the result.

| Organ |  | Cu | Fe | Mn | Zn |
|---|---|---|---|---|---|
| Content (mg/kg) | Root | 13.65 ± 4.56a | 962 ± 390a | 32.99 ± 10.12b | 25.88 ± 10.90b |
|  | Stem | 14.73 ± 4.79a | 578 ± 299b | 27.94 ± 12.03b | 32.38 ± 13.49c |
|  | Leaf | 15.28 ± 4.28a | 605 ± 518b | 65.89 ± 19.45a | 39.88 ± 10.75a |
| Yield (mg/m$^2$) | Root | 2.95 ± 1.72A | 234 ± 190A | 8.34 ± 7.16A | 6.46 ± 5.77A |
|  | Stem | 2.28 ± 1.01B | 93 ± 70B | 4.49 ± 2.96B | 5.04 ± 2.81AB |
|  | Leaf | 1.46 ± 1.09C | 57 ± 61B | 6.31 ± 4.68AB | 3.84 ± 2.99B |

Note:
The significant difference was determined by one-way ANOVA, and it was represented as the different lowercase letters on the contents, and as the different capital letters on the yields ($P < 0.05$).

## Age-effect on the contents and yields of Cu, Fe, Mn and Zn in whole lucerne

The whole biomass of older lucerne plants (11 and 8 years old) was significantly higher than the biomass in younger ages (5, 4 and 1 years old, $P < 0.05$, Fig. 1B). There were no significant difference of Cu and Fe contents and yields of the whole lucerne on the age-effect ($P > 0.05$, Table 4). The Mn and Zn contents of 1 year old was significantly higher than the other four ages ($P < 0.05$), while the Mn and Zn yields decreased first, then increased with the increased ages, and the lowest yields were shown in the 5 year old samples.

## Organ-effect on the contents and yields of Cu, Fe, Mn and Zn in lucerne

The biomass was highest in the root, higher in the stem and lowest in the leaf ($P < 0.05$, Fig. 1C).

Cu content varied indistinctively among leaf, stem and root ($P > 0.05$, Table 5). Root Fe content was significantly higher than in the leaf and stem ($P < 0.05$). Leaf Mn and Zn

**Table 6 The average contents and average yields of Cu, Fe, Mn and Zn in all cuts and ages.**

|  | Cu | Fe | Mn | Zn |
|---|---|---|---|---|
| Content (mg/kg) | 14.15 ± 4.35b | 738 ± 282a | 37.15 ± 9.23b | 30.81 ± 9.59b |
| Yield (mg/m²) | 6.69 ± 3.01B | 385 ± 268A | 19.14 ± 11.87B | 15.34 ± 8.57B |

**Note:**
The average ± standard deviation was shown as the results. The significant difference was determined by one-way ANOVA, it was represented as the different lowercase letters on the contents, and as the different capital letters on the yields ($P < 0.05$).

**Table 7 Significant correlations among the biomass, contents and yields of Cu, Fe, Mn and Zn.**

|  |  | Content (mg/kg) | | | | Biomass (g/m²) | Yield (mg/m²) | | | |
|---|---|---|---|---|---|---|---|---|---|---|
|  |  | Cu | Fe | Mn | Zn |  | Cu | Fe | Mn | Zn |
| Content (mg/kg) | Cu | 1 | ns | ns | 0.626** | −0.347* | 0.420** | ns | ns | ns |
|  | Fe |  | 1 | 0.871** | 0.452** | ns | ns | 0.682** | 0.539** | 0.482** |
|  | Mn |  |  | 1 | 0.440** | 0.307* | ns | 0.658** | 0.623** | 0.513** |
|  | Zn |  |  |  | 1 | ns | 0.418** | ns | ns | 0.537** |
| Biomass (g/m²) |  |  |  |  |  | 1 | 0.654** | 0.860** | 0.922** | 0.827** |
| Yield (mg/m²) | Cu |  |  |  |  |  | 1 | 0.605** | 0.637** | 0.772** |
|  | Fe |  |  |  |  |  |  | 1 | 0.965** | 0.868** |
|  | Mn |  |  |  |  |  |  |  | 1 | 0.881** |
|  | Zn |  |  |  |  |  |  |  |  | 1 |

**Note:**
The data of all stand ages and cuts were used for analyzing the correlations. "ns" was represented insignificance ($P > 0.05$). The asterisk (*) and double asterisks (**) represent significant correlations at $P < 0.05$ and $P < 0.01$, respectively.

contents were significantly higher compared to the root and stem ($P < 0.05$). The yields of Cu, Fe, Mn and Zn were significantly higher in the root than in the stem and leaf ($P < 0.05$).

## Compared to the average contents and average yields of Cu, Fe, Mn and Zn in whole lucerne

The average contents and average yields were both shown as the Fe > Mn > Zn > Cu (Table 6). And the average contents and average yields of Fe were significantly higher than the other elements ($P < 0.05$).

## Correlations among the biomass, contents and yields of Cu, Fe, Mn, and Zn in whole lucerne

The Cu content increased with the Zn content, and it positively contributed to the accumulations for the yields of Cu, but it decreased with the increased biomass of the whole plant (Table 7). The Fe content increased with the Mn and Zn contents, and it positively contributed to the accumulations on yields of Fe, Mn and Zn. The Mn content increased with the increased Zn content, and it positively correlated with biomass and the yields of Fe, Mn and Zn. The Zn content positively correlated with the yields of Cu and Zn. Biomass contributed to the yields of Cu, Fe, Mn, and Zn. The yields of Cu, Fe, Mn and Zn are shown as the pairwise correlations.

## DISCUSSION

### Age-effect on Cu, Fe, Zn and Mn in lucerne

The mean contents of microelements differed intrinsically in different ages and were related with element types in this study. But a common characteristic that the highest contents of Cu, Fe, Mn and Zn were found in the 1 year old lucerne samples in this study. And *Kuang et al. (2007)* also found the contents of Fe, Mn and Zn were higher in younger needles of *Pinus massoniana*. However, the yields of Cu, Fe, Mn and Zn were highest in the older and/or younger year old stands. Those depended on the contents of the four microelements and the biomass of lucerne. Positive correlations were also found between yields and contents or biomass of lucerne in this study. Those were similar as the carbon and nitrogen accumulation of forest plants (*Hooker & Compton, 2003*; *Yang & Luo, 2011*). The "reserve emerged" root increased the yield of Cu, Fe, Mn and Zn as the changing trend of lucerne biomass, although lucerne would be cut the aboveground parts may act as forage many times in one year. Furthermore, the results differ from what was reported in garlic chives where the leaves were removed frequently in an experiment on three year old samples (*Żurawik, Jadczak & Żurawik, 2013*). They found leaf Mn content of garlic chives was lowest in the 2 year old samples than in the 1 and 3 year old samples, Cu and Zn contents decreased, and Fe contents increased with stand age. The difference may be due to the different species or organs that were sampled.

### Cut-effect on Cu, Fe, Zn and Mn in lucerne

Shoots of lucerne could be cut and removed frequently each year (*Maadi, Sakinejad & Seyedmohammadi, 2013*). These may lead to changes in element status and cycling in the lucerne ecosystem (*Wang et al., 2014*, *2015*, *2019*), including the contents of lucerne Cu, Fe, Mn and Zn. And the microelements were higher in third cut than first and second cuts in this study, which similarly changed the lucerne biomass of the three cuts (Fig. 1A). However the biomass of lucerne would decrease with the increased cuts in previous reports (*Tyrolová & Výborná, 2008*; *Du et al., 2013*). These results differed as the measured biomass was discrepant as whole plant or aboveground. Although the aboveground of lucerne was cut frequently, the root contributed more biomass for the whole plant (Fig. 1C), which would accumulate the microelements more in this study. This is similar to the elemental accumulation in forest plants (*Hooker & Compton, 2003*; *Yang & Luo, 2011*). Furthermore, the changeable trend of biomass and the contents of Cu, Fe, Mn and Zn positively correlated with the yields of Cu, Fe, Mn and Zn in this study. These led to the highest yields of the four microelements being in the third cut.

### Organ-effect on Cu, Fe, Zn and Mn in lucerne

Stem and leaf of lucerne were often studied as the utilization for forage in previous research (*Tyrolová & Výborná, 2008*; *Vystavna et al., 2019*). The different percentages from whole plants contributed to the variable nutrient contents (*Tyrolová & Výborná, 2008*). The microelement as the mineral element is also changeable in different organs of lucerne (*Marković et al., 2009*), which has been proven in this study. They all found the contents of leaf Cu, Fe, Mn and Zn were higher than in the stem, and even higher than in the root

(except the Fe contents), which suggested the leaves needed more nutritional support for physiological activity, i.e., photosynthesis etc. *Adeniyi & Olatunji (2019)* also proved leaf Cu, Fe, Mn and Zn were higher than in the stem. Meanwhile, the root contributed more biomass than stem or leaf for the whole plant, the yields of Cu, Zn, Mn and Fe showed the similar results as the biomass of the three organs of lucerne. This is also proven by the correlations between the biomass and the yields of the four microelements in this study.

## Correlations among the Cu, Fe, Mn, and Zn in whole lucerne

Plant Cu, Fe, Mn and Zn were derived from soil pool, but their contents and yields were variety (*Behera et al., 2015*; *Azuma, Tomioka & Takenaka, 2016*). Previous literature reported that exogenous Zn application would enhance Zn content, but did not influence the contents of Cu, Mn and Fe in maize (*Behera et al., 2015*). This might mean that there were non-significant correlations between the contents of Zn and the contents of Cu, Mn and Fe in maize. And the similar results were also found in *Polygonatum cyrtonema* Hua (*Chen et al., 2020*). But now significantly positive correlations were found in lucerne. Perhaps because the species or fertilizer application led to the differences. Furthermore, Zn application led to more yield of maize, and then enhanced the total Zn, Mn, Cu and Fe uptake (*Behera et al., 2015*). This is approved by the positive correlations between biomass and yields of Cu, Zn, Fe and Mn in this study.

## CONCLUSIONS

The experiment found that the highest contents of Cu, Fe, Mn and Zn were found in the 1 year old among the five ages, at the 3rd cut, and in the leaf. But the higher yields of Cu, Fe, Mn and Zn were found in the older ages (11 and 8 years old), at the third cut, and in the root. Furthermore, the contents of microelements and plant biomass contributed significantly positive roles to the yields of microelements in lucerne. Therefore, it is suggested that lucerne microelements are age-, cut- and organ-specific. And the accumulation of microelements would increase even with the frequent mowing of lucerne.

### Funding

This work was supported by the Forage Industrial Innovation Team, Shandong Modern Agricultural Industrial and Technical System, China (SDAIT-23-10), the Natural Science Foundation of Shandong Province (ZR2020QC184), the Key Innovation Project on Agriculture Applied Technology from The Key Innovation Project on Agriculture Applied Technology from Shandong Province Government (2018–2020). Research and Demonstration of the Integrated Ecological Cycle for Planting and Feeding Model, the Sheep Industrial Innovation Team, Shandong Modern Agricultural Industrial and Technical System, China (SDAIT-10-14), and the Technology Innovation Team for Animal Behavior & Welfare and Health Breeding. The funders had no role in study design, data collection and analysis, decision to publish, or preparation of the manuscript.

## Grant Disclosures

The following grant information was disclosed by the authors:

Forage Industrial Innovation Team, Shandong Modern Agricultural Industrial and Technical System, China: SDAIT-23-10.

Natural Science Foundation of Shandong Province: ZR2020QC184.

Shandong Province Government: 2018–2020.

Sheep Industrial Innovation Team, Shandong Modern Agricultural Industrial and Technical System, China: SDAIT-10-14.

## Competing Interests

The authors declare that they have no competing interests.

## Author Contributions

- Zhennan Wang conceived and designed the experiments, performed the experiments, analyzed the data, prepared figures and/or tables, and approved the final draft.
- Yizhao Shen analyzed the data, authored or reviewed drafts of the paper, guaranteed the accuracy of English, and approved the final draft.
- Chongliang Bi analyzed the data, prepared figures and/or tables, and approved the final draft.
- Mirielle Pauline analyzed the data, authored or reviewed drafts of the paper, guaranteed the accuracy of English, and approved the final draft.
- Qingping Zhang performed the experiments, prepared figures and/or tables, and approved the final draft.
- Shenjin Lv analyzed the data, prepared figures and/or tables, and approved the final draft.
- Huimin Yang conceived and designed the experiments, analyzed the data, authored or reviewed drafts of the paper, and approved the final draft.
- Yan Yang analyzed the data, authored or reviewed drafts of the paper, and approved the final draft.

## Data Availability

   Raw data is available as a Supplemental File.

## Supplemental Information

Supplemental information for this article can be found online at http://dx.doi.org/10.7717/peerj.11188#supplemental-information.

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
