# Peer review of "Contents and yields of copper, iron, manganese and zinc would be affected by lucerne age and cut"

_PeerJ, doi:10.7717/peerj.11188_

## Round 0.1 · original submission · Major Revisions

Major Revision is required according to the reviewers' reports. Improve the quality of your paper according to the standard of PeerJ.

Reviewer 1 ·

Basic reporting

no comment

Experimental design

in the experimental design, authors defined the age, cut of the lucerne, but these definitions should be detailed introduction as the results.

Validity of the findings

no comment

Additional comments

This manuscript presented the changing microelements of lucerne with stand ages, cuts and organs. Authors verified the characteristic of the changing factors. These results have certain significance to understand the growth law of lucerne. But there were also some questions as follows. Thus, the manuscript can be accept after correct these questions.
1,in the experimental design, authors defined the age, cut of the lucerne, but these definitions should be detailed introduction as the results.
2,Please correct the number of decimal places for uniform data retention in same line of tables.
3,The headings were too detailed in result, please combine and redescribe them.

Reviewer 2 ·

Basic reporting

no comments.

Experimental design

no comments.

Validity of the findings

no comments.

Additional comments

Line33:”cutting period” must be given the specific growth stage, maybe”early flowering stage”?
Line 37: or should be changed as “and”.
Line 27 and 43: Unified microelements or trace element.
Line 82: “plateau” changed as “Plateau”.
Line 83:“480mm to 660mm”changed as “480~660 mm”.
Line 91: the cutting period? When or which stage?
Line 102-103: “ml” changed as “mL”.
Line 109: Three-way analysis? The correct description maybe Multivariate analysis of variance analysis?

Reviewer 3 ·

Basic reporting

- The English language could be improved to ensure that an international reader can clearly understand your text. Some examples where the language could be improved include but not limit to:
 [line 29-30] It would be helpful to rephrase this sentence to allow readers to better understand it.
 [line 30-31] What do you mean by “stand age”? It would be helpful to rephrase this sentence to allow readers to better understand it.
 [line 33-34] It could be changed to “the contents and yields of Cu, Fe, Mn and Zn in lucerne”.
 [line 35-36] It’s a little confusing when you use “respectively” here. It would be helpful to rephrase this sentence to allow readers to better understand it.
 [line 36-37] The same “respectively” issue, it would be helpful to rephrase this sentence to allow readers to better understand it.
 [line 46] “basal”  “basic” Is that what you mean to say?
 [line 48] It’s a little bit confusing about the use of “and” and “but” in this sentence. It would be helpful to rephrase this sentence to allow readers to better understand it.
 [line 58] “Microelement as macroelement” could be changed to “Similar to microelement, microelement…”
 [line 60] delete the “and” at the end of this line
 [line 61] It’s a little bit confusing about the use of “As well” in this sentence. It would be helpful to rephrase this sentence to allow readers to better understand it.
 [line 66] The “ha” in the phrase “32 million ha” could be replaced by its full name “hectare”, because this is the first time to use it. Then you can choose to use abbreviation later.
 [line 94] It’s a little bit confusing about the use of “secondly” in this sentence. It would be helpful to rephrase this sentence to allow readers to better understand it.
 [line 99] “digestive tube”  “digestion tube”
 [line 100] It would be helpful to mention the concentration of nitric acid you used.
 [line 101] It would be helpful to describe more details about your chemical analysis. For example, you could provide more details on your digestion method and parameters you used (e.g., temperature, power, use gradient program or not, etc).
 [line 103] “solution of beaker”  “sample solution”
 [line 104] It’s a little bit confusing about the use of “at a constant volume” in this sentence. Do you mean you measured and recorded the volume?
 [line 104] “ice” needs to be replaced by its full name. And please detail your analytical method and parameters when you use this instrument. It would be very helpful for the readers to reproduce your work.
 [line 211] “most positive correlations were found” seems a little bit vague. It would be better to restate your results and conclusions more clearly.

Experimental design

In this paper, the research question was relatively well-defined and meaningful. Both experimental and statistical investigation have been performed. However, it would be more helpful to add more description and information to help readers to better understand on the experiments and results.

Validity of the findings

The results are valid and useful.

Additional comments

It would be helpful to update your references and cite more recent research to replace relatively old ones.

Reviewer 4 ·

Basic reporting

Please see my comments in "General comments for the author"

Experimental design

Please see my comments in "General comments for the author"

Validity of the findings

Please see my comments in "General comments for the author"

Additional comments

The article presents a study of contents and yields of copper, iron, manganese, and zinc
in lucerne of different ages and cuts, as well as the differences among organs. Statistical analyses were also employed to confirm the alteration. Overall, the language is fair, and the manuscript is generally well-organized.

Major scientific comments:
1. In the “Introduction” section, the motivation of this study is unclear. It seems that the authors did this study because “the elements play important roles, so we want to know the alterations”. The authors had a small paragraph (line 58-64) discussing the relationship between microelements and plant growth and the health of livestock. This might be a meaningful reason that drives this study. However, the authors just huddled thru with limited examples. It is suggested to include more details/examples in this paragraph to interest readers. The authors should introduce more on the relationship of microelements and plant growth/animal health.
Similarly, in the “Conclusions” section, authors should give some perspective on how the current study benefits, but not just give a summary of the results.
2. In the Methods section, line 98-106, the authors employed a digestion&spectrometer-based method to analyze the chemicals. Is this method previously reported? If so, please cite it. Otherwise, authors need to do some method validations, e.g. linear range, LOD/LOQ, accuracy/precision. It is important to know these parameters since some of the results differed from the literature, which was discussed in the “Discussion” section. Note if the method is not accurate for the analytes, it will also lead to differences.
Line 105-106, the authors claimed, “Lastly, the contents were calculated by a formula.” Is this “formula” well-known? Or is it created by the standard curve made in house? Please cite or show the formula if it is known. Otherwise, if it is built by a standard curve, authors need to indicate the standard chemicals they used.
3. Significant correlations among the biomass, contents and yields of Cu, Fe, Mn and Zn were shown in Table 7. It is suggested to move this paragraph to the “Discussions” section. Moreover, there is no single reference in this paragraph when the authors discussed the correlations among elements. Instead, the authors used ambiguous sentences “ The Fe content increased with the Mn and Zn contents, and it positively contributed to the accumulations on yields of Fe, Mn and Zn.” I assume there are plenty of studies regarding relationships of element up-taking in the literature. The authors should find and cite them instead of just guessing.

Minor comments

4. In Table 4-6, lowercase letters “a” and “b” were shown. Their meanings were not clearly annotated. I assume they are referred as “ The significant difference was determined by one-way ANOVA, and it was represented as the different lowercase letters (P<0.05).” Additionally, the authors need to clarify what the difference is between two lowercase letters. Probably, one for p<0.01 while another for p<0.05?

---

## Round 0.2 · Minor Revisions

The manuscript is revised and now improved a lot. I checked and found that the number of references is less (just 27), and updated references 2017-2020 are missing. Please check the manuscript once again, check English and update references, thanks.

---

## Round 0.3 · Minor Revisions

Dear Editors

I checked the revised manuscript and the responses of the authors to comments. The manuscript is now looking very improved and close to being suitable for publication in PeerJ.

Before that, the Section Editor had a couple of concerns that you should clarify in a minor revision:

1. The manuscript is a bit confusing in that the experiments were done over eight years ago and some of the experimental protocols are cited with a less than eight time-period stamp - indicating that the data were collected more recently. Please explain this disagreement.

2. Please confirm whether this work has been published previously, possibly in a language other than English.

---

## Round 0.4 · accepted · Accept

The manuscript is revised according to the comments of reviewers.